# Evolving Cognitive Dysfunction in Children with Neurologically Stable Opsoclonus–Myoclonus Syndrome

**DOI:** 10.3390/children7090103

**Published:** 2020-08-19

**Authors:** En Lin Goh, Kate Scarff, Stephanie Satariano, Ming Lim, Geetha Anand

**Affiliations:** 1Oxford University Clinical Academic Graduate School, Medical Sciences Division, University of Oxford, Oxford OX3 9DU, UK; enlin.goh@doctors.org.uk; 2Department of Paediatrics, Children’s Hospital Oxford, Oxford University Hospitals NHS Foundation Trust, Oxford OX3 9DU, UK; geetha.anand@ouh.nhs.uk; 3Department of Clinical Neuropsychology, Oxford Psychological Medicine Centre, Oxford OX3 9DU, UK; kate.scarff@ouh.nhs.uk; 4Children’s Neuroscience Centre, Evelina London Children’s Hospital, Guy’s and St Thomas’ NHS Trust, London SE1 7EH, UK; stephanie.satariano@gstt.nhs.uk; 5Department of Women and Children’s Health, Faculty of Life Sciences, King’s College, London SE1 7EH, UK

**Keywords:** opsoclonus–myoclonus syndrome, neurodevelopment, cognitive dysfunction

## Abstract

Cognitive and acquired neurodevelopmental deficits have been reported in children with opsoclonus–myoclonus syndrome (OMS) and are known to be associated with more severe and relapsing disease course. However, there is a paucity of data regarding cognitive dysfunction in children with stable neurological disease. We report three children with OMS and evolving cognitive dysfunction in the context of a mild disease course. The children’s ages at disease onset were between 17 and 35 months and they were followed up for 4–10 years. Neuroblastoma was identified in one child. OMS severity scores ranged between 8 and 12/15 at presentation. They underwent immunotherapy and all were in remission by 7 months (range 4–13 months), with treatment maintained for 1 year. One child remained relapse-free, while two others had one clinical relapse each and were immunotherapy-responsive again. In all cases, evolving cognitive dysfunction was reported despite being in remission and stable off treatment for a median of 20 months (range of 12–31 months; two OMS scores of 0/15 and one of 2/15). In children with OMS who have completed treatment and have made full or near full neurological recovery, concerns remain regarding long-term outcome in terms of future learning and cognitive development.

## 1. Introduction

Paediatric opsoclonus–myoclonus syndrome (OMS) is a rare condition with an annual incidence of one in five million in the United Kingdom (UK) [1]. This condition is characterised by rapid, involuntary eye movements, myoclonic jerks, ataxia, sleep disturbances and, in the long-term, behavioural and learning difficulties. Furthermore, there is a strong association between OMS and neuroblastoma [2]. Children with OMS display functionally active auto-antibodies, pro-inflammatory changes in the cytokine network and response to immunosuppressive treatment, thus suggesting underlying auto-immune phenomena [3].

Studies investigating long-term cognitive sequelae have reported progressive and pervasive neurodevelopmental deficits following the onset of the condition [4,5,6,7,8]. These neurodevelopmental sequelae are most significant in children in whom the disease runs a chronic course, with multiple relapses and prolonged treatment [6]. Notably, the negative correlation of functional status with age at testing have raised concerns that OMS is a progressive encephalopathy [5,6]. Several mechanisms by which this occurs have been postulated, including damage to cerebellar–cortical circuits at onset of the disease that becomes more apparent with time or persistent ongoing low grade inflammation [9,10].

There continues to be a paucity of information regarding cognitive dysfunction in children with stable neurological disease. Furthermore, this is compounded by the lack of availability of data from serial neuropsychological testing. Here, we report serial neuropsychological assessments of three children with OMS demonstrating evolving cognitive dysfunction with milder disease course.

## 2. Materials and Methods

After observing an unexpected evolution of cognitive dysfunction in our index case (see later Case 1), we reviewed the cases of OMS presenting to two tertiary neuroscience centres to identify further cases of children with this progressive deterioration of cognitive dysfunction during serial neuropsychological testing despite disease stability, defined as being in disease remission and not necessitating further immunotherapy. A medical case note review was performed to outline the OMS disease course. The following neuropsychological outcomes were assessed as part of standard clinical care: Full-scale intelligence quotient (FSIQ) as measured by the Wechsler Preschool and Primary Scale of Intelligence Third/Fourth Edition (WPPSI-III/IV) or Wechsler Intelligence Scale for Children Fourth/Fifth Edition (WISC-IV/V); verbal intelligence quotient (VIQ) relating to general verbal intellectual ability as measured by the WPPSI-III Verbal IQ Index or WPPSI-IV/WISC-IV/V Verbal Comprehension Index; and performance intelligence quotient (PIQ) relating to general non-verbal intellectual ability as measured by the WPPSI-III Performance IQ Index, WPPSI-IV/WISC-V Fluid Reasoning Index or WISC-IV Perceptual Reasoning Index. Scores reported are standard scores which have a mean of 100 and standard deviation of 15. Where necessary, substitution and proration of subtest/index scores were obtained in line with test administration guidelines to achieve the index scores. Other cognitive domains tested were dictated by clinical need and age of the child but typically included a measure of attention, memory, language and academic attainment. Consent was obtained from all patients for reporting of clinical and investigative data.

## 3. Results

Three children (two girls) diagnosed with OMS between 17 and 35 months were followed up for 4–10 years. The detailed clinical history of each of the cases is provided in the Appendix A, while the key clinical and investigative features and disease course are summarised in Table 1.

Neuroblastoma was identified in one child. OMS severity scores ranged between 8 and 12/15 at presentation. Patients underwent immunotherapy with a widely used European regime currently adopted in European OMS trial (see NCT01868269) [11]. Notably, all presentation predated the trial study period. All patients were in remission by 7 months (range 4–13 months), with treatment maintained for 1 year. One child remained relapse-free, while the remaining two children had one clinical relapse each, which was immunotherapy-responsive. Of note, one child (Case 1) experienced two cerebellar infarctions at 5 years and 1 month and 7 years and 1 month, respectively, and was found to have vertebral artery dissection. In all cases, evolving cognitive dysfunction was reported despite being in remission and stable off treatment for 20 months (see Figure 1, range of 12–31 months; 2 OMS scores of 0/15 and one of 2/15). In two of the cases (Cases 2 and 3), repeat neuroimaging was performed following final cognitive assessment which were normal

Serial cognitive assessment scores for each child are illustrated in Figure 1 and reported in Table 2. At the index cognitive assessment, FSIQ, VIQ and PIQ scores ranged from 81 to 100, 95 to 108 and 101 to 104, respectively. At the final cognitive assessment, decline in FSIQ, VIQ and PIQ scores were noted, which ranged from 52 to 81, 53 to 81 and 69 to 82, respectively. Significant deficits were observed in terms of attention, processing speed, visuospatial skills, and language, which were not initially apparent at initial assessment.

## 4. Discussion

In the present case series, we report evolving cognitive dysfunction occurring in children with OMS who have a milder disease and long after completion of treatment. It is evident that these children continued to develop but with a widening gap in comparison with peers, therefore suggesting impairment of future learning and cognitive development. Numerous studies have documented the association between OMS and neurodevelopmental deficits [4,7,8]. Mitchell and colleagues reported a relationship between age of testing and degree of cognitive deficit where the youngest, most recently treated children were functioning at nearly normal levels, whereas deficits were more prominent in older children [5].

A subsequent study of 19 children by the same group raised the question as to whether OMS should be considered a progressive encephalopathy due to the long-term cognitive decline noted [6]. In both studies, children with a chronic, relapsing disease course were noted to have the most striking cognitive deficits, while children with a monophasic course displayed a more benign prognosis [5,6]. Similarly, a recent paper by Gorman et al. showed that the number of relapses and final OMS severity score were strong predictors of final FSIQ [12]. In our cohort, one child was relapse-free following initial treatment, while two children experienced one relapse each, which were responsive to immunotherapy, representing children with a milder disease course. Our findings indicate that a residual risk persists even in children with a milder disease course that is not ameliorated despite treatment. It must be noted that one child experienced cerebellar and minor thalamic infarctions and was shown to have sequential vertebral artery dissection. It is plausible that the cognitive testing outcomes observed in this child could be confounded by the sequelae of cerebellar infarction and, thus, not solely attributable to OMS.

Several hypotheses have been postulated to explain the neurodevelopmental changes. Damage to cerebellar–cortical circuits at the onset of disease have been implicated. We have observed that cerebellar grey matter is significantly reduced in patients with OMS, most prominently in the vermis and flocculonodular lobe along with reduced cortical thickness, indicating extra-cerebellar involvement. The severity of cerebellar atrophy is related to persistent symptomatology [9]. Alternatively, persistent low-grade inflammation may play a role. Pranzatelli and colleagues reported high levels of activated B lymphocytes in the cerebrospinal fluid of children with OMS that persisted years after onset [10]. Follow-up neuro-imaging in two of our cases did not reveal evidence of any inflammatory changes or progressive atrophy suggestive of non-OMS related pathology. It is well documented that the vast majority of clinically interpreted MRI scans are normal in both the acute and late stages of OMS, and therefore, the normal MRI does not exclude the low-grade inflammation hypothesis. Group-wise comparisons of OMS patients versus normal subjects using voxel based morphometry have suggested lower volume of the vermis but the MRI scans used as part of our work-up were not of sufficient resolution to evaluate these changes [9].

Impairment of complex attention, receptive language issues and poor naming that typifies chronic brain inflammation as reported in paediatric-onset multiple sclerosis are not what is seen in our cases [13]. By contrast, our cases display cognitive deficits observed in children who have sustained significant acquired brain injury in infancy and to an extent those following pre-term births. In the case of brain injury in infancy, longitudinal data suggest that the child often “grows into” their deficits over time with “new” impairments becoming apparent on serial testing as the child gets older [14]. In particular, lower-order skills such as simple language and visual skills show a relatively good recovery, but complex skills, which only become apparent as a child gets older, remain restricted. While children born pre-term are at risk of delays in cognitive function, there is some evidence to show that this difference corrects by 24 months initially but becomes apparent again at a primary school age [15,16,17,18]. Hence, it is plausible that in the initial stages of disease, compensatory mechanisms resulting in brain functional reorganisation similar to brain “scaffolding” may enable improved cognitive function, but the limits are reached with increased age [19,20].

This case series illustrates that there can be very substantial cognitive deficits that emerge over time in a small group of these patients. The pattern and degree of cognitive deficit is reminiscent of significant early damage to the developing central nervous system, as is also seen in cases with acquired brain injury or whole brain irradiation for brain tumour [21]. Similar cognitive developmental trajectories can also be seen, but to a more limited extent, in children of premature birth [18]. As with early acquired brain injury, it may be that lower-order skills are relatively well-preserved, but more complex skills which only become apparent as child gets older remain restricted, leading to generalised cognitive dysfunction. It may also be that, as with children born prematurely, compensatory mechanisms resulting in functional reorganisation may enable lower-level cognitive function, but that the limits are reached with increased age. Further work needs to be done to understand the underlying mechanisms of this evolving cognitive dysfunction and to elucidate the risk factors for some children having such poor neurocognitive outcomes.

## 5. Conclusions

In children with stable OMS who have completed treatment, concerns remain regarding long-term outcome in terms of future learning and cognitive development. Our findings highlight the need for early and regular assessment of cognitive functioning. Larger, longitudinal studies are necessary to identify trends and characteristics in OMS patients with a mild disease course. Functional serial neuroimaging in the subgroup of OMS children with poor neurocognitive outcome will help to ascertain whether they as a group show long-term progressive sequalae of perturbations to cerebellar–cortical circuits.

## Figures and Tables

**Figure 1 children-07-00103-f001:**
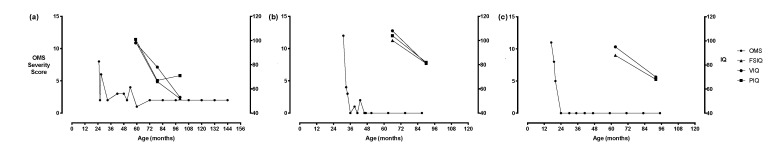
Relationship between clinical course and intellectual changes for (**a**) Case 1, (**b**) Case 2 and (**c**) Case 3.

**Table 1 children-07-00103-t001:** Summary of clinical features.

Case	Gender	Age at Onset	Age at Diagnosis	Clinical Features at Presentation	OMS Score at Diagnosis	Presence of Neuroblastoma	Treatment	Response	Age at Last Follow-Up
1	Male	23 months	25 months	Unsteady gait, abnormal eye movements, opsoclonus, titubation, intention tremor and ataxia	8/15	No	Prednisolone IVIG	Good	13 years
2	Female	30 months	34 months	Unsteady gait, vomiting, opsoclonus and regression in language	12/15	No	Dexamethasone	Good	7 years 6 months
3	Female	17 months	17 months	Limb tremors, ataxia, loss of lower limb and truncal control and loss of speech	11/15	Yes	Surgical resection of neuroblastoma Dexamethasone Cyclophosphamide	Good	7 years 8 months

**Table 2 children-07-00103-t002:** Summary of cognitive skills. Key: ‘−‘: age-appropriate (within 1 standard deviation of mean), ‘+’: moderate impairment (1–2 standard deviations below normative mean), ‘++’: severe impairment (≥2 standard deviations below normative mean).

Case	Age	Intellectual Functioning	Other Domains Tested
1	4 years 11 months	FSIQ: 99 VIQ: 98 PIQ: 101	Attention (selective): – Visuospatial skills: ++ Language (receptive): – Language (expressive): –
	6 years 7 months	FSIQ: 66 VIQ: 78 PIQ: 67	Processing speed: ++ Language (receptive): – Language (expressive): –
	8 years 4 months	FSIQ: 5 VIQ: 53 PIQ: 71	Attention (working memory): ++ Processing speed: ++ Visuospatial skills: ++ Language (receptive): ++ Language (expressive): ++ Academic (reading): ++ Academic (spelling): ++
2	5 years 6 months	FSIQ: 100 VIQ: 108 PIQ: 104	Attention (sustained): – Attention (working memory): – Processing speed: – Visuospatial skills: + Memory (new learning): – Memory (delayed recall): –
	7 years 6 months	FSIQ: 81 VIQ: 81 PIQ: 82	Attention (sustained): ++ Attention (working memory): – Processing speed: – Visuospatial skills: + Language (receptive): – Memory (new learning): – Memory (delayed recall): – Academic (reading): – Academic (spelling): –
3	5 years 3 months	FSIQ: 81 VIQ: 95	**Not tested**
	7 years 8 months	FSIQ: 68 VIQ: 70 PIQ: 69	Attention (sustained): ++ Processing speed: ++ Language (receptive): ++ Language (expressive): – Memory (new learning): – Memory (delayed recall): – Academic (reading): – Academic (spelling): –

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
