# Peer review of "Evolving Cognitive Dysfunction in Children with Neurologically Stable Opsoclonus–Myoclonus Syndrome"

_children, 2020, doi:10.3390/children7090103_

Round 1
Reviewer 1 Report
children-890946
The manuscript reports on 3 children who had OMS, which was relatively easily controlled without aggressive immunotherapy. Specifically, treatment included only corticosteroids in one, corticosteroids plus IVIG in one, and corticosteroids plus cyclophosphamide in one. None received B-cell depletion with rituximab or other agents. All three had striking drops in scores on a variety of developmental and cognitive testing over several years, with notable changes in behavior and learning reported by parents and teachers. The bulk of the case material is in the supplementary materials. The cases are interesting and all had OMS, but case 1 (the index case) has progressive deficits that cannot be confidently attributed solely to OMS. The authors report that case 1 had an unexplained small thalamic infarction, then developed cerebellar infarctions and had two apparently spontaneous dissections of vertebral arteries. The timing of first noted infarction and dissection was shortly after the first apparently developmental testing. The second assessment showing significant drops in IQ was about 1½ years later. No etiology for the repeated strokes and vertebral dissections is mentioned. For example, was the child evaluated for a vasculopathy, either inflammatory or genetic? Was there any indication of connective tissue disorder such as Ehler-Danlos Syndrome or Marfan’s Syndrome? Was there a family history of stroke or vasculopathy? Was there evaluation for thrombophilia? While the serial test scores could plausibly be due to late effects of OMS, or even ongoing low grade neuroinflammation, it is not fully convincing in view cerebellar infarction.
The authors make several statements that are not entirely supported by the OMS literature. Although reference 3 (Blaes) does, in fact, report active autoantibodies, this has never been replicated by any other lab to date, despite extensive searches for auto-antibodies. The authors also argue that late MRIs which do not show ongoing neuroinflammation is evidence for lack of OMS activity. (line 125-6) However, even during the acute phase, brain MRI, even with gadolinium contrast, generally do not show signs of neuroinflammation in OMS. Inflammatory changes on neuroimaging have never been reported during late OMS exacerbations. A few authors have reported post-OMS patients with midline cerebellar atrophy, but the vast majority of clinically interpreted MRI scans on OMS survivors are normal. There are reports of group-wise comparisons of OMS patients versus normal subjects using voxel based morphometry that have suggested lower volume of the vermis, but this is not generally evident on clinically interpreted scans.
Lastly one very minor suggestion: In abstract line 21, add “at onset”. It should be “The children’s ages at onset were between 17 and 35 months”
Author Response
The cases are interesting and all had OMS, but case 1 (the index case) has progressive deficits that cannot be confidently attributed solely to OMS. The authors report that case 1 had an unexplained small thalamic infarction, then developed cerebellar infarctions and had two apparently spontaneous dissections of vertebral arteries. The timing of first noted infarction and dissection was shortly after the first apparently developmental testing. The second assessment showing significant drops in IQ was about 1½ years later. No etiology for the repeated strokes and vertebral dissections is mentioned. For example, was the child evaluated for a vasculopathy, either inflammatory or genetic? Was there any indication of connective tissue disorder such as Ehler-Danlos Syndrome or Marfan’s Syndrome? Was there a family history of stroke or vasculopathy? Was there evaluation for thrombophilia? While the serial test scores could plausibly be due to late effects of OMS, or even ongoing low grade neuroinflammation, it is not fully convincing in view cerebellar infarction.
Thank you for highlighting this important point. We had given careful consideration to this confounding factor and had concluded that despite it, it was worth including case 1 in our paper given that cognitive plateauing was observed in three other cases (two cases have been included in this manuscript; one could not be due to difficulty in contacting the family to obtain consent). We have included this in the discussion and more details regarding his investigations and family history in the supplement.
Lines 135-138 (Discussion) – “It must be noted that one child experienced cerebellar and minor thalamic infarctions and was shown to have sequential vertebral artery dissection. It is plausible that the cognitive outcomes observed could be confounded by the sequelae of cerebellar infarction and thus, not solely attributable to OMS.”
Lines 273-276 (Supplement) – “There was no family history of stroke. There was no clinical evidence of connective tissue disorders such as Marfan’s or Ehlers Danlos syndrome. Thrombophilia screen was negative. There was no clinical evidence of vasculopathy. Anti-nuclear antibodies (ANA) and anti-neutrophil cytoplasmic antibodies (ANCA) screening were negative.”
The authors make several statements that are not entirely supported by the OMS literature. Although reference 3 (Blaes) does, in fact, report active autoantibodies, this has never been replicated by any other lab to date, despite extensive searches for auto-antibodies.
We acknowledge that the reference we used is not the best. The Oxford Team recently identified autoantibodies to GluD2 patients with OMS, and concluded that they are potentially pathogenic. We have updated the reference accordingly.
Lines 417-419 (References) – “Berridge, G.; Menassa, D.A.; Moloney, T.; Waters, P.J.; Welding, I.; Thomsen, S.; Zuberi, S.; Fischer, R.; Aricescu, A.R.; Pike, M.; et al. Glutamate receptor δ2 serum antibodies in pediatric opsoclonus myoclonus ataxia syndrome. Neurology 2018, 91, e714–e723.”
The authors also argue that late MRIs which do not show ongoing neuroinflammation is evidence for lack of OMS activity. (line 125-6) However, even during the acute phase, brain MRI, even with gadolinium contrast, generally do not show signs of neuroinflammation in OMS. Inflammatory changes on neuroimaging have never been reported during late OMS exacerbations. A few authors have reported post-OMS patients with midline cerebellar atrophy, but the vast majority of clinically interpreted MRI scans on OMS survivors are normal. There are reports of group-wise comparisons of OMS patients versus normal subjects using voxel based morphometry that have suggested lower volume of the vermis, but this is not generally evident on clinically interpreted scans.
Thank you for your insightful points. We fully agree and have amended the manuscript. We have also referenced this. (Anand, G.; Bridge, H.; Rackstraw, P.; Chekroud, A.M.; Yong, J.; Stagg, C.J.; Pike, M. Cerebellar and cortical abnormalities in paediatric opsoclonus‐myoclonus syndrome. Dev. Med. Child Neurol. 2015, 57, 265–272.)
Lines 159-165 (Discussion) – “Follow-up neuro-imaging in two of these children did not reveal any evidence of inflammatory changes or progressive atrophy suggestive of non-OMS related pathology. It is well documented that vast majority of clinically interpreted MRI scans are normal in both the acute and late stages of OMS and therefore, the normal MRI does not exclude the low-grade inflammation hypothesis. There are reports of group-wise comparisons of OMS patients versus normal subjects using voxel based morphometry that have suggested lower volume of the vermis but the MRI scans used as part of our work-up were not of sufficient resolution to evaluate these changes.”
Lastly one very minor suggestion: In abstract line 21, add “at onset”. It should be “The children’s ages at onset were between 17 and 35 months”
Thank you for pointing this out. We have amended this.

Reviewer 2 Report
The authors report neuropsychological findings in the long-term course after successfull treatment of childhood opsoclonus-myoclonus syndrome. Over the course of some years they report a progressive, apparently linear decline of FSIQ, VIQ and PIQ by at the end roughly 20 IQ points. They discuss the possible etiology and stress the necessity of continuous follow-up in these children to detect the neuropsychological deficits early to enable adequate intervention.
This is an important observation, especially concerning the long-term decline of IQ. The question of an ongoing inflammatory process versus a static encephalopathy remains open. The cited comparability with the IQ course in former prematures is in favour of a static condition with a decreased potential of further development. I can add to this discussion that the same phenomenon is also found in children after wholebrain-radiotherapy for brain tumors or leukemia, and in some unfavourable early childhood epilepsies. As a child neurologist I would try to find out if the patients "have lost skills that they had already acquired" (which would be an argument for a progressive encephalopathy), or if the decline of IQ is merely to "slower than normal development" (which would be in favour of a static encephalopathy and "pseudodementia").
I have two further comments to improve the manuscript:
- It is ok that the authors in their results section present only the most important and summarized findings, and present the detailed histories as a supplement. However, in case 1 the repeated cerebellar infarctions are very important and should be included in the main presentation. Cerebellar pathology has been reported to be associated with cognitive function - and this is even the main topic of this report.
- Table 2 has to be re-edited because at present it is not possible to recognize how the "other domains tested" belong to the time course and IQ data. Furthermore, I do not understand what the signs (+++, -) after the neuropsychological domains mean.
Author Response
This is an important observation, especially concerning the long-term decline of IQ. The question of an ongoing inflammatory process versus a static encephalopathy remains open. The cited comparability with the IQ course in former prematures is in favour of a static condition with a decreased potential of further development. I can add to this discussion that the same phenomenon is also found in children after wholebrain-radiotherapy for brain tumors or leukemia, and in some unfavourable early childhood epilepsies. As a child neurologist I would try to find out if the patients "have lost skills that they had already acquired" (which would be an argument for a progressive encephalopathy), or if the decline of IQ is merely to "slower than normal development" (which would be in favour of a static encephalopathy and "pseudodementia").
We have made the following clarification in our ‘Discussion’ to address this comment.
Lines 179-190 (Discussion) – “This case series illustrates that there can be very substantial cognitive deficits that emerge over time in a small group of these patients. The pattern and degree of cognitive deficit is reminiscent of significant early damage to the developing central nervous system, as is also seen in cases with acquired brain injury or whole brain irradiation for brain tumour. Similar cognitive developmental trajectories can also be seen, but to a more limited extent, in children of premature birth. As with early acquired brain injury, it may be that lower-order skills are relatively well-preserved but more complex skills which only become apparent as child gets older remain restricted, leading to generalised cognitive dysfunction. It may also be that, as with children born prematurely, compensatory mechanisms resulting in functional reorganisation may enable lower-level cognitive function, but that the limits are reached with increased age. Further work needs to be done to understand the underlying mechanisms of this evolving cognitive dysfunction and to elucidate the risk factors for some children having such poor neurocognitive outcomes.”
I have two further comments to improve the manuscript:
- It is ok that the authors in their results section present only the most important and summarized findings and present the detailed histories as a supplement. However, in case 1 the repeated cerebellar infarctions are very important and should be included in the main presentation. Cerebellar pathology has been reported to be associated with cognitive function - and this is even the main topic of this report.
We have now included this in the main presentation.
Lines 84-86 (Results) – “Of note, one child (Case 1) experienced two cerebellar infarctions at 5 years and 1 month and 7 years and 1 month, respectively.”
We have also elaborated the Discussion to highlight this aspect of Case 1.
Lines 135-138 (Discussion) – “It must be noted that one child experienced cerebellar and minor thalamic infarctions and was shown to have sequential vertebral artery dissection. It is plausible that the cognitive testing outcomes observed could be confounded by the sequelae of cerebellar infarction and thus, not solely attributable to OMS.”
- Table 2 has to be re-edited because at present it is not possible to recognize how the "other domains tested" belong to the time course and IQ data. Furthermore, I do not understand what the signs (+++, -) after the neuropsychological domains mean.
Thank you for this comment. We have edited Table 2 to include a key on Lines 101-104 to clarify this further. Please note that for Case 3, no other domains were tested at the first assessment. Additionally, we have included the following statement in the Methods section.
Lines 69-70 – “Other cognitive domains tested were dictated by clinical need and age of the child, but typically included a measure of attention, memory, language and academic attainment.”

Round 2
Reviewer 1 Report
The authors have made the changes I suggested in the first review.